# Dried Blood Spots (DBS): A suitable alternative to using whole blood samples for diagnostic testing of visceral leishmaniasis in the post-elimination era

**Prakash Ghosh**[1]◦*, **Rajashree Chowdhury**[1]◦, **Md. Abu Rahat**[1]◦, **Faria Hossain**[1], **Nur E Arpha**[2], **Mojca Kristan**[3], **Matthew Higgins**[3], **Ahmed Abd El Wahed**[4], **Yasuyuki Goto**[5], **M. M. Towhidul Islam**[6], **Susana Campino**[3], **Mary Cameron**[3], **Malcom S. Duthie**[7], **Rashidul Haque**[8], **Dinesh Mondal**[1]*

**1** Nutirition and Clinical Services Division, International Centre for Diarrhoeal Disease Research, Banladesh, Dhaka, Bangladesh, **2** Genetic Engineering and Biotechnology, BRAC University, Dhaka, Bangladesh, **3** London School of Hygiene and Tropical Medicine, University of London, London, United Kingdom, **4** Institute of Animal Hygiene and Veterinary Public Health, University of Leipzig, Leipzig, Germany, **5** Laboratory of Molecular Immunology, Department of Animal Resource Sciences, Graduate School of Agricultural and Life Sciences, The University of Tokyo, Tokyo, Japan, **6** Department of Biochemistry and Molecular Biology, University of Dhaka, Dhaka, Bangladesh, **7** HDT Bio, Seattle, Washington, United States of America, **8** Emerging Infections and Parasitology Laboratory, International Centre for Diarrhoeal Disease Research, Bangladesh, Dhaka, Bangladesh

◦ These authors contributed equally to this work.
* prakash.ghosh@icddrb.org (PG); din63d@icddrb.org (DM)

**Data Availability Statement:** The authors confirm that all data underlying the findings are fully available without restriction. All relevant data are

## Abstract

### Background

Serum or whole blood collection, processing, transport and storage still present significant challenges in low resource settings where mass surveillance is required to sustain disease elimination. Therefore, in this study, we explored the diagnostic efficacy of dried blood spots (DBS) as a minimally invasive and potentially cost-effective alternative sampling technique to whole blood sampling procedures for subsequent detection of *Leishmania donovani* antibodies or DNA.

### Methodology and principal findings

Archived serum, DNA samples from whole blood of visceral leishmaniasis (VL) cases and healthy controls, and DBS from corresponding cases and controls, were used. Both molecular and serological assays were optimized to detect *L. donovani* antibodies or DNA in DBS elute and results were compared against those obtained with whole blood. Serological assays (both rK28 ELISA and rK39 ELISA) of DBS samples showed sensitivity and specificity of 100% and had excellent agreement with results from whole blood samples (kappa value ranged from 0.98–1). Bland-Altman analysis of OD values from rK28-ELISA with DBS elute and patients' serum showed an excellent agreement (ICC = 0.9) whereas a good agreement (ICC = 0.8) was observed in the case of rK39-ELISA. However, qPCR and RPA

within the paper and its Supporting Information files.

**Funding:** The funding was granted by the Swedish International Development Cooperation Agency (Sida,GR-01455), Sweden and Rainy Day Grant Fund under 'Young investigator's award', International Centre for Diarrhoeal Disease Research, Bangladesh (icddr,b) to PG. The authors alone are responsible for the views expressed in this manuscript. The funders had no role in study design, data collection and analysis, decision to publish, or preparation of the manuscript.

**Competing interests:** The authors have declared that no competing interests exist.

of DBS samples had a diminished sensitivity of 76% and 68%, respectively, and poor agreement was observed with the whole blood samples.

## Conclusion

Our results demonstrate that DBS offer excellent diagnostic efficiency for serological assays and represent a viable alternative to whole blood sampling procedures.

## Author summary

Since the inception of the National Kala-azar Elimination Program (NKEP) in 2005, visceral leishmaniasis (VL) cases have declined significantly in Bangladesh. Bangladesh has achieved the elimination targets set by the WHO and continuous strides have been enforced by the government and non-government organizations in the present maintenance phase. To sustain the outcomes of the NKEP, move towards a zero-transmission goal and subvert the endgame challenges, an effective surveillance strategy is a primary focus. As part of this strategy, we examined whether using dried blood spots (DBS), rather than whole blood samples as currently recommended, could serve as a minimally invasive sampling technique prior to subsequent diagnostic tests. We determined the accuracy of molecular and serological assays using DBS samples compared with whole blood samples. While using DBS, excellent diagnostic accuracy was observed for serological assays (rK28 ELISA and rK39 ELISA), whereas sensitivity and specificity using molecular assays for DNA detection (qPCR and RPA) declined in comparison to whole blood. Given the low cost and field feasibility of DBS, we believe this simple sample collection technique coupled with serology-based detection of *L. donovani* meets the requirements for mass surveillance towards detecting residual VL transmission.

## Introduction

Despite significant successes, efforts towards elimination of Neglected Tropical Diseases (NTDs) are being continued in the endemic tropics in accordance with the revised NTD Roadmap for 2021–2030 [1]. With perpetuating efforts, since its inception in 2005, the regional kala-azar elimination program has been on a trajectory towards eliminating the disease from the Indian subcontinent. The primary tangible achievement of the National Kala-azar Elimination Program (NKEP) in Bangladesh includes the reduction of cases down to 1 per 10,000 individuals at upazila level in VL endemic districts [2]. Towards cementing the achievements of the program, the maintenance phase has started which requires cost-effective strategies for long-term transmission control and disease surveillance as per the recommendations of VL experts [3]. However, considering the lack of investments in NTDs, and the unavailability of resources, implementation of a cost-effective diagnostic tool should be the most pragmatic option for surveillance through mass screening.

To date, rK39 antigen based rapid diagnostic tests (RDTs) have been considered the most pragmatic option for detection of *L. donovani* antibodies in serum samples from patients with VL or VL-associated complications. Although popular, most RDTs have operational limitations because, according to typical manufacturer directions, RDTs should be performed using serum. In our experience the first difficulty with this approach is that children, and even adults, often refuse to give venous blood required for serum extraction. Secondly, where the use of

centrifuges or microfuges is not common at the point-of-care, this necessitates blood collection, transportation, and fractionation. Therefore, a traditional blood sampling procedure coupled with rK39 RDT is not ideal for mass surveillance in resource poor endemic settings.

Dried blood spot (DBS) is a minimally invasive technique that involves a very small volume of finger-pricked blood being added on a filter paper-based substrate, followed by drying prior to analysis. This simple blood collection and preservation technique was first introduced by Guthrie *et al.*, in 1963 to screen congenital phenylketonuria in neonates [4]. With proper extraction of analytes from the DBS a large number of bioanalytical procedures, such as chromatography, mass spectrometry, DNA analysis and enzyme immunoassays have been performed with DBS across several research fields [4].

The logistical ease of transport and storage has enabled DBS samples to become a universal tool for public health data collection and integration. In the last few decades protocols for analyte extraction from DBS to enable use in molecular and serological assays throughput have improved significantly. Accordingly, the costs per assay have dropped to levels that allow for viral, bacterial and parasite detection and control even in resource-poor settings [5–6]. Considering the multifarious advantages of DBS, this alternative approach of collecting capillary blood by finger prick followed by preservation on filter paper could streamline mass screening efforts to a greater extent than collection of venous blood. Following collection and transportation, downstream serological and molecular detection methods for leishmaniasis can then be performed at central facilities.

Reports on the diagnostic merits of filter paper blood or DBS samples for diagnosis of visceral leishmaniasis are scarce even though a few studies have shown promising diagnostic efficacy of DBS for cutaneous leishmaniasis [7–14]. Therefore, as an important validation step prior to implementation for mass screening, it is imperative to determine the diagnostic efficacy of DBS samples using serological or molecular assays for detecting a past or present VL infection. It is equally important to assess whether the efficacy of DBS samples is as good as whole blood for diagnosing VL. In our previous study, we showed excellent diagnostic efficacy of direct agglutination test (DAT) performed with DBS samples [15]. However, due to various factors such as operational difficulties, reagent expense and the longer assay period, DAT is not suitable for mass screening. In this study we determined the suitability of DBS samples for use in serological rK39 and rK28 ELISAs as well as in qPCR and using isothermal recombinase polymerase amplification (RPA) molecular assays.

## Methods and materials

### Ethics statement

This study was approved by the Institutional Review Board (IRB) and Ethical Review Committee (ERC) of International Centre for Diarrhoeal Disease Research, Bangladesh (icddr,b), research protocol number PR-20040. Written informed consent was obtained from individuals and/or minors' legal guardians for use of their collected samples for VL research.

### Study design, study sites and participant characteristics

This was a laboratory-based study with a case-control design involving participant enrollment and laboratory activities. Enrollment of study participants and sample collection were conducted at Surja Kanta Kala-azar Research Centre (SKKRC) situated in Mymensingh district, a highly endemic zone for VL that accounts for more than half of the total VL cases in Bangladesh. Laboratory activities were performed in the Emerging Infections and Parasitology Laboratory, icddr,b, Dhaka. All participants were enrolled between March 2018 to October 2018. In total, 50 clinically confirmed VL patients and 50 control participants were included in this

study. All VL patients were diagnosed clinically by the hospital physician in accordance with the national guideline for VL [16]. VL cases were defined as individuals from an endemic area with fever > 2 weeks, splenomegaly and positive rK39 RDT test. All VL cases enrolled in this study were also positive using DAT. Healthy individuals from the same region, with no previous VL history, nor any symptoms of severe, acute, or chronic illness, and a negative rK39 RDT/DAT test were enrolled as controls or endemic healthy controls (EHC). Following standard measures, VL patients were enrolled by the study physician and their hospital admission, clinical management and treatment was set at SKKRC. According to the national guidelines, each VL patient was treated with a single dose intravenous infusion of 10 mg/kg Liposomal amphotericin B (LAmB) and followed for 12 months after treatment to assess their response and potential development of VL associated post-treatment complications [16].

## Clinical specimens collection

5 milliliters of blood was collected from all study participants at SKKRC using aseptic precautions. Subsequently, blood was centrifuged at 3,500 rpm for 5 minutes for separation of a serum specimen. Additionally, to prepare DBS, 50µL of whole blood was impregnated on to Whatman FTA cards (Sigma-Aldrich) and allowed to dry. For laboratory analysis, specimens were transported to Emerging Infections and Parasitology Laboratory, icddr,b while maintaining a proper cold chain (4˚C).

## Molecular detection methods for diagnosis of VL

**Template preparation from clinical specimens.**   DNA was extracted from 200 μL heparin-treated whole blood and from three 5mm punched-out discs from DBS by DNeasy blood and tissue DNA extraction kits following the manufacturer's instructions (QIAGEN, Hilden, Germany). DNA extracted from whole blood and DBS was eluted into 200 μL and 150 μL of elution buffer provided with the kits, respectively, and stored at –20˚C until performing downstream tests.

**qPCR.**   Following a previously established protocol, targeting the conserved REPL repeats specific for *L. donovani* and *L. infantum*, qPCR was performed with extracted template DNA from whole blood and DBS (S1 Table) [2,17]. In brief, to prepare a 20 μL reaction mix, 9 μL template DNA, 10 μL of TaqMan Gene Expression Master Mix (Applied Biosystems) and 1 μL pre-ordered Taqman primer-probe mix (Applied Biosystems) were combined. For amplification, Bio-Rad CFX96 iCycler system was utilized. The following were the amplification conditions: 10 min at 95˚C, followed by 15 seconds at 95˚C and 1 min at 60˚C for 45 cycles. In each run, a standard curve was created with 10 ng—1 fg of parasite DNA isolated from *in vitro* cultured promastigotes (*L. donovani* MHOM/IN/80/DD8) corresponding to 10,000–0.1 parasites per reaction. One reaction with nuclease-free water was also included in each run as a negative control. A sample was considered as negative with a cycle threshold (Ct) > 40. Each sample was evaluated in duplicate, and an additional run was made in case of an inconclusive result.

**Recombinase polymerase amplification (RPA) assay.**   The RPA assay was carried out with template DNA extracted from blood and DBS using a protocol described elsewhere (S2 Table) [18]. In summary, a TwistAmp exo kit (Product code#TAEXO02KIT, TwistAmp exo kits, TwistDx, Cambridge, UK) was used to perform the assay. In a tube 420 nM of RPA primer, 120 nM of RPA Probe, and 1× rehydration buffer were mixed to prepare a 50 μL reaction volume which was added to the RPA lyophilized pellet. 14 mM of Mg acetate were added to the tube lids. Template DNA was added to the tubes, which were then closed and mixed thoroughly. The tubes were immediately placed into the tube scanner (Twista, TwistDx, Cambridge, UK) and incubated for 15 minutes at 42˚C. At 20 s intervals, the emitted fluorescence

signals were measured. Signal was interpreted utilizing a combined threshold and first derivative analysis. It took approximately 20 minutes to complete the RPA assay.

## Serology-based diagnostic methods

**Sample preparation for serological assays.**   Serum samples and DBS were used to perform ELISA. To collect the DBS extract, two 5 mm punched-out circles from the center of each spot were placed in eppendorf tubes containing 150 μl of extraction buffer (PBS containing 0.01% Tween 20) and incubated at room temperature with gentle shaking for 1.5 hours. After centrifugation, the supernatants were used to perform ELISA assay.

**rK39 and rK28 ELISA.**   Antibody capture ELISA was carried out using the procedures as described previously [19–20]. Briefly, rK39 and rK28 antigens in phosphate-buffered saline (PBS), pH 7.4 (25ng/well for both serum and DBS) were coated on 96-well flat bottom microtiter plates (Corning, NY) and incubated overnight at 4˚C. To avoid non-specific reactivity, plates were gently washed three times with wash buffer (PBS containing 0.05% Tween 20; pH 7.4) and then incubated for 2 hours at 37˚C with blocking buffer (PBS containing 15 BSA; pH 7.4). Following washing steps, 50 μl of 1:200 times diluted serum and DBS samples in dilution fluid (DF: PBS containing 0.1% BSA and 0.05% Tween 20) were added to each well and incubated for 1 h at 37˚C. Subsequently, the plates were rinsed five times with wash buffer, and 100 μl of peroxidase-conjugated rabbit anti-human IgG (Jackson Immnuoresearch, PA, USA) diluted 1:5,000 times in DF was added to each well and incubated for 1 h at 37˚C. After washing five times, each well was filled with 100 μl of TMB substrate (Sigma-Aldrich, Missouri, USA), and the plate was left in the dark for 20 min. To stop the reaction, 1N $H_2SO_4$ was added to each well and then the optical density (OD) of each well was measured at 450nm in a microreader plate (Synergy 2 plate reader, Biotek, Winooski, VT).

**Statistical analysis.**   The clinical sensitivity and specificity of the assays were assessed compared to the VL case definition according to the national guideline as a gold standard. For rK39 and rK28 ELISA, a receiver-operator characteristic curve (ROC) was created to determine all probable combinations of sensitivity and specificity, and an optimal cut-off was selected that clearly distinguish VL and controls at 95% confidence intervals. The value of the area under the curve derived from ROC analysis was considered as the measure used to assess the diagnostic accuracy of each antigen in this study. Test performance of each method was investigated by calculating sensitivity and specificity with 95% confidence intervals using standard statistical formulae. The Bland-Altman plot with 95% confidence interval was used to evaluate the agreement between OD values of ELISA assay obtained with serum and DBS. 95% LOA was calculated by the average difference between serum and DBS ± 1.96SD (upper and lower LOA). The narrower the 95% LOA, the better the agreement. In addition, the reliability of the DBS method was tested by intraclass correlation coefficient (ICC); if the ICCs were closer to 1, the reliability was higher. McNemar's test was performed to determine the concordance and discordance among the investigated molecular and serological methods. Further, Mann–Whitney's U test was performed to measure the differences between CT value and TT value from qPCR and RPA assays respectively with both whole blood and DBS extracted DNA. Spearman's correlation was used to assess the correlation between the assays using different samples. Inter-assay agreement was measured by determining Cohen's kappa coefficient (k) values. According to Landis and Koch, the values of Cohen's k coefficients were interpreted as follows: 1.00–081: excellent; 080–061: good; 060–041: moderate; 040–021: poor; and 020–000: insignificant agreement [21]. SPSS 20, Graphpad Prism 8.0.2 and R version 3.4.0 (R Foundation, Vienna, Austria) software were used to conduct all statistical analyses.

**Table 1. Clinical and demographic characteristics of the study populations.**

| Variables | VL cases (N = 50) | Endemic Controls (N = 50) |
|---|---|---|
| Sex | | |
| Male, n (%) | 28 (56%) | 29 (58%) |
| Female, n (%) | 22 (44%) | 21 (42%) |
| Age (years) Mean (SD) | 31.50 (14.99) | 27.64 (14.21) |
| History of VL, n (%) | 23 (46%) | 0 (0%) |
| History of PKDL | 2 (4%) | 0 (0%) |
| Fever $\geq$ 14 days, n (%) | 50 (100%) | 0 (0%) |
| Duration of fever in days Mean (SD) | 109.3 (110) | NA |
| Splenomegaly, n (%) | 50 (100%) | 0 (0%) |
| Hepatomegaly, n (%) | 14 (28%) | 0 (0%) |
| Pancytopenia, n (%) | 28 (56%) | 0 (0%) |
| Positive in rK39 RDT, n (%) | 50 (100%) | 0 (0%) |
| Positive DAT, n (%) | 50 (100%) | 0 (0%) |

## Results

### Demographic statistics of participants

The clinical and demographic details of the study participants are presented in (Table 1). Among 50 clinically confirmed recruited VL patients, 46% were relapse cases and 2 patients had previous history of post-kala-azar dermal leishmaniasis (PKDL). Approximately sixty percent of the study participants were male. The mean age of VL patients and healthy controls were 31.50 and 27.64 years respectively. All of the VL patients had fever for more than two weeks and exhibited splenomegaly, while hepatomegaly was found in 28% of the VL patients. All of the VL patients enrolled were found to be responsive to the treatment, were clinically cured within 6 months and reported no VL-related clinical issues within 12 months after treatment.

### Diagnostic performance of molecular assays

In this study, we evaluated a previously established qPCR and RPA assays for detecting *L. donovani* DNA in clinical samples from VL patients. As expected, a higher sensitivity was obtained for the qPCR assay, 94.00% (95% CI: 83.45%-98.75%), with DNA extracted from whole blood (WB-qPCR) compared to 76.00% (95% CI: 61.83%-86.94%) when utilizing DNA extracted from DBS samples (DBS-qPCR) (Table 2). While utilizing the WB DNA and DBS DNA, the sensitivity of the WB-RPA and DBS-RPA assay was 92.00% (95% CI: 80.77%-97.78%) and 68.00% (95% CI: 53.30%-80.48%) respectively (Table 2). Both the molecular assays achieved 100% specificity with the DNA extracted from whole blood and DBS. The median parasite load of whole blood qPCR was 304.44 (IQR: 126.11–697.22) parasite/mL and the median parasite load for FP qPCR was 256.9 (IQR:111.11–407.99) parasite/mL.

**Table 2. Diagnostic performance of molecular methods using different types of samples.**

| Methods | Sample Type | Sensitivity (95% CI) (n/N) | Specificity (95% CI) (n/N) |
|---|---|---|---|
| qPCR | Whole Blood | 94.00% (83.45%-98.75%) (47/50) | 100.00% (92.89%-100.00%) (50/50) |
| | Dried Blood Spot | 76.00% (61.83%-86.94%) (38/50) | 100.00% (92.89%-100.00%) (50/50) |
| RPA | Whole Blood | 92.00% (80.77%-97.78%) (46/50) | 100.00% (92.89%-100.00%) (50/50) |
| | Dried Blood Spot | 68.00% (53.30%-80.48%) (34/50) | 100.00% (92.89%-100.00%) (50/50) |

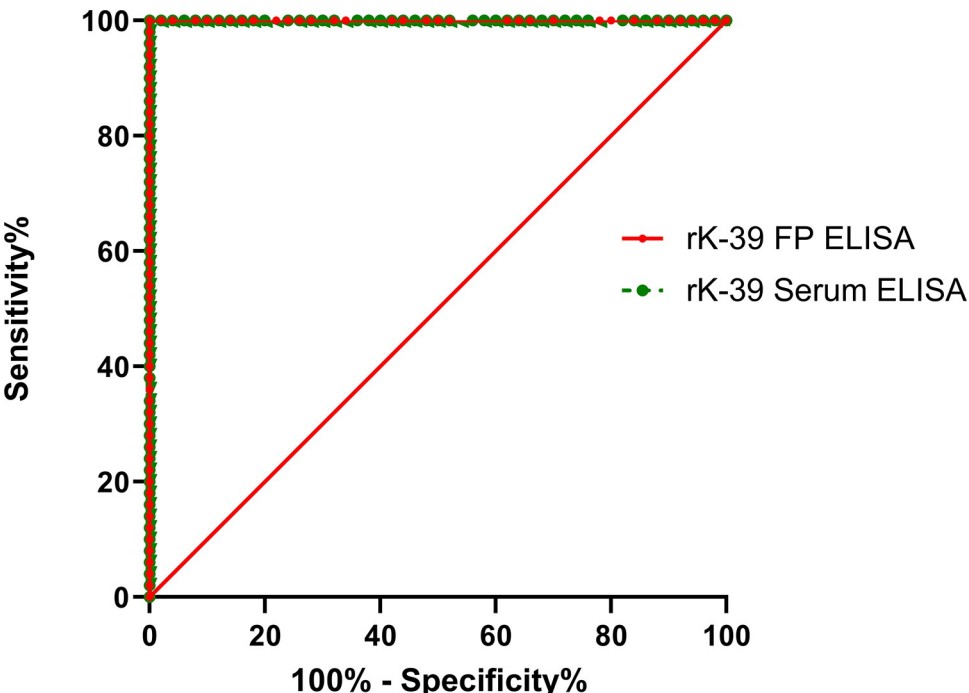

**Fig 1. Multiple ROC Curve constructed from the OD (450nm) values obtained from conducting rK39 ELISA in serum and DBS samples.**

## Diagnostic performance of serology-based methods

To detect anti-rK39 and anti-rK28 antibodies, ELISA was performed with paired serum and DBS samples. After evaluating the absorbance readings at 450nm, the cut-off values were measured by ROC analysis (Figs 1 and 2). For serum samples (S), the cut-off values of rK39 and rK28 (S-rK39 and S-rK28) were 0.291 and 1.002, whereas for DBS samples (DBS) the cut-off values of rK39 and rK28 (DBS-rK39 and DBS-rK28) were 0.044 and 0.048 respectively (Table 3). According to the cut-off values obtained from ROC curve, S-rK39, DBS-rK39 and DBS-rK28 ELISA showed an absolute sensitivity (95% CI: 92.89%-100.00%), while S-rK28 ELISA presented a marginally lower sensitivity of 98.00% (95% CI: 92.89%-100.00%) (Table 4). Each of the antigens achieved a specificity of 100.00% (95% CI: 92.89%-100.00%) (Table 4). The ROC curve indicated a high degree of diagnostic accuracy for rK39 and rK28 ELISA using both serum and DBS specimens, as the value of AUC ranged from 0.997–1.00 (Table 3).

## Impact of blood sampling methods on molecular and serological assays

Following the performance of RPA and qPCR with whole blood and DBS, a congruency in the diagnostic efficiency of the defined methods was observed. WB-qPCR assay showed an excellent agreement with WB-RPA and DBS-RPA (Fig 3). Similarly, the DBS-qPCR assay represented an excellent agreement with WB-RPA and DBS-RPA, while a good agreement was observed with WB-qPCR. Thus, a good agreement was demonstrated between the qPCR and RPA while conducting the assays with either whole blood or DBS. There was a significant difference (p <0.0001) between the Ct values of WB-qPCR and DBS-qPCR assay (Fig 4). Likewise, WB-RPA and DBS-RPA reported a significant difference (p <0.0019) on TT values (Fig 5).

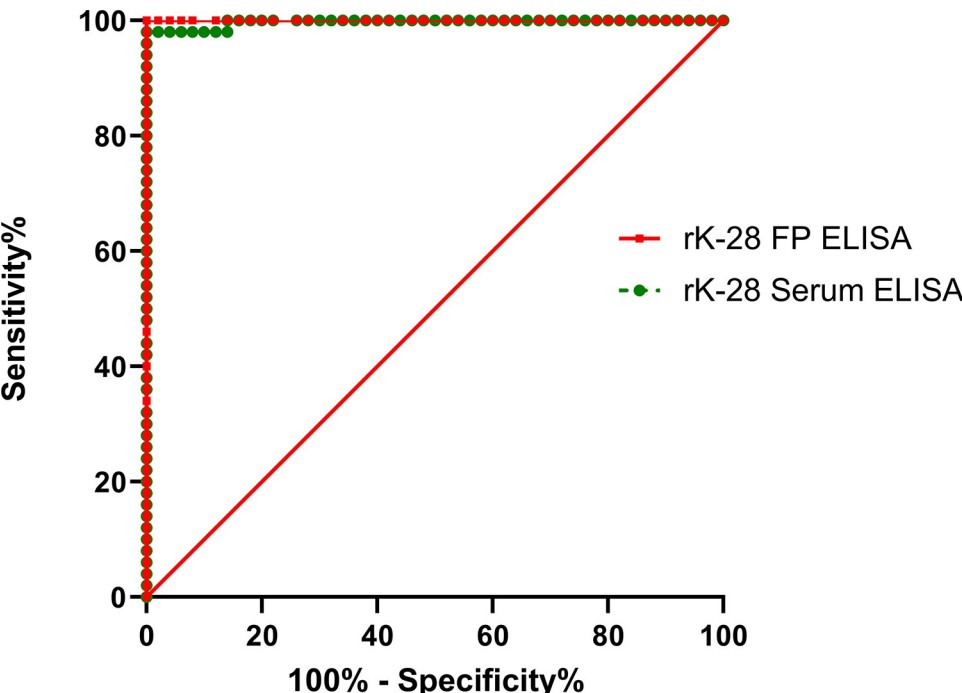

**Fig 2. Multiple ROC Curve generated from the OD (450nm) values obtained from rK28 ELISA in serum and DBS samples.**

**Table 3. The AUC and cut-off OD values of ELISA methods performed with rK39 and rK28 recombinant antigens using serum and DBS samples.**

| ELISA | Sample Type | Cut-off OD | Area under the curve |
|---|---|---|---|
| rK39 | Serum | 0.29100 | 1.000 |
| | Dried Blood Spot | 0.04375 | 1.000 |
| rK28 | Serum | 1.00200 | 0.997 |
| | Dried Blood Spot | 0.04750 | 1.000 |

**Table 4. Diagnostic performance of serology-based methods using different types of samples.**

| Methods | Sample Type | Sensitivity (95% CI) (n/N) | Specificity (95% CI) (n/N) |
|---|---|---|---|
| rK28 ELISA | Serum | 98.00% (89.35%-99.95%) (49/50) | 100.00% (92.89%-100.00%) (50/50) |
| | Dried Blood Spot | 100.00% (92.89%-100.00%) (50/50) | 100.00% (92.89%-100.00%) (50/50) |
| rK39 ELISA | Serum | 100.00% (92.89%-100.00%) (50/50) | 100.00% (92.89%-100.00%) (50/50) |
| | Dried Blood Spot | 100.00% (92.89%-100.00%) (50/50) | 100.00% (92.89%-100.00%) (50/50) |

As expected, excellent agreement was found among serology-based methods with different sampling procedures with a kappa value of ≥0.98. Additionally, in Bland-Altman analysis, an excellent agreement (ICC = 0.9) was observed between S-rK28 and DBS-rK28 ELISA assay whereas S-rK39 and DBS-rK39 ELISA assay showed a good agreement (ICC = 0.8). The OD values for serum and DBS samples obtained from VL patients, in both rK39 and rK28 ELISA assay, spanned within the 95% limits of agreement in Bland-Altman plot (Fig 6). From the Bland-Altman analysis, the difference between OD values obtained with serum and DBS tend to decrease with the increase of the average OD value in both rK28 and rK39 ELISA (Fig 6).

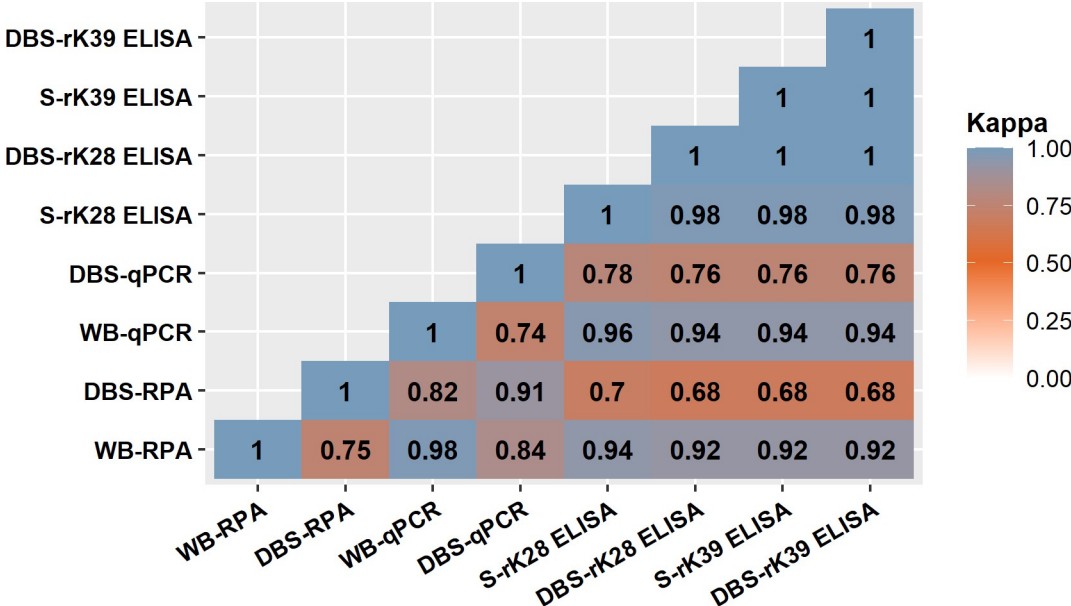

**Fig 3. Measure of agreement presented as Cohens Kappa coefficients between molecular and serological methods employing different clinical specimens (WB–whole blood; DBS–dried blood spot; S–serum).**

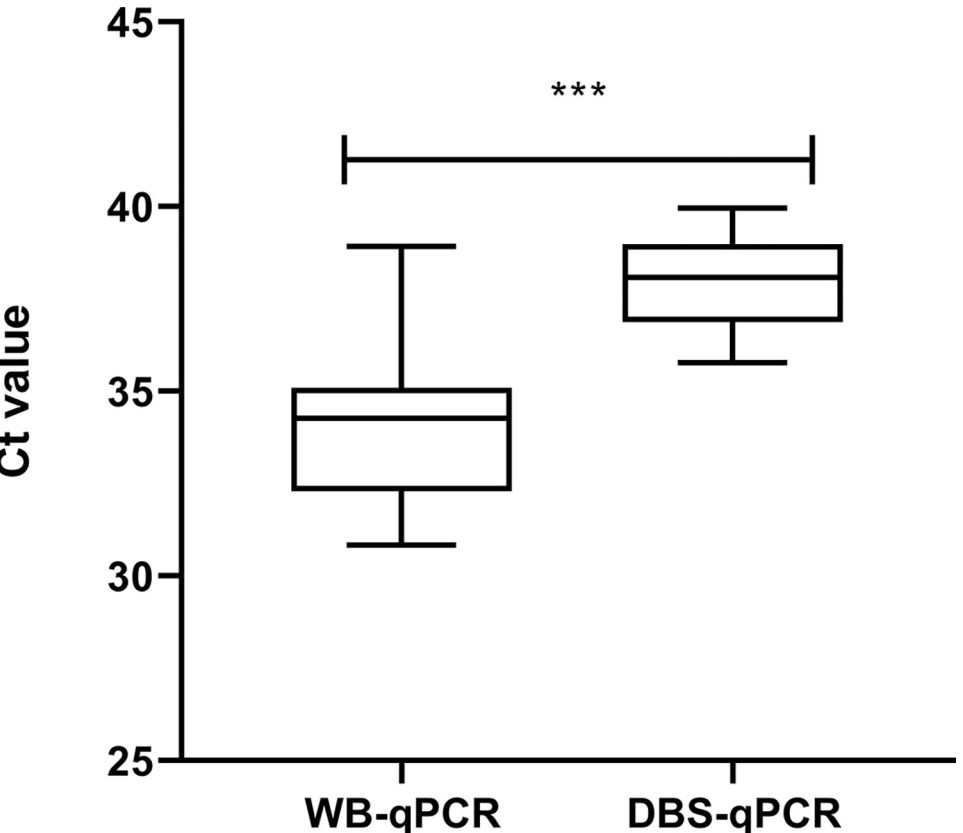

**Fig 4. Differences in Ct values by qPCR assay between whole blood (WB) and dried blood spot (DBS) extracted DNA.**

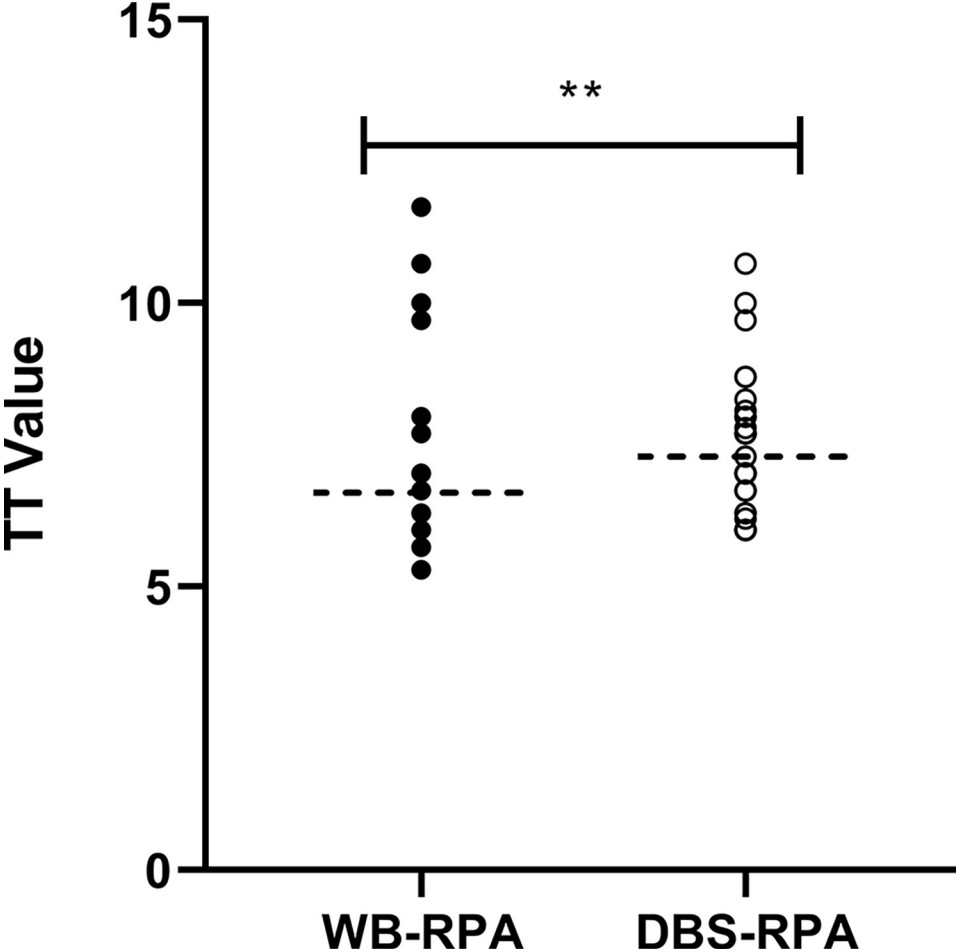

**Fig 5. Differences in TT values by RPA assay between whole blood (WB) and dried blood spot (DBS) extracted DNA.**

Furthermore, a strong positive correlation (r = 0.8856) was observed between S-rK28 and DBS-rK28 ELISA methods and the finding was statistically significant (p<0.0001). In parallel, a substantial positive correlation (r = 0.8101) was observed between the S-rK39 and DBS-rK39 ELISA assays which was statistically significant (p<0.0001) as well. In comparison with the molecular assays, serology-based methods showed excellent and good agreement for WB samples and DBS respectively.

## Discussion

With the implementation of a broad holistic NKEP, Bangladesh has already achieved its target by decreasing the new case detection rate to less than 1 in 10,000 among the population at the upazila level by 2017 [22]. To maintain and further progress towards achieving the zero-transmission goal and permanent elimination of the disease, during this post elimination phase, periodic surveillance to gauge the residual infection levels should be continued. Although several urine-based surveillance tools have been devised, broad-scale validation of such tools has not been established. Therefore, as a means to circumvent the endgame challenges at the present time, developing an effective and sustainable mass surveillance strategy around the existing

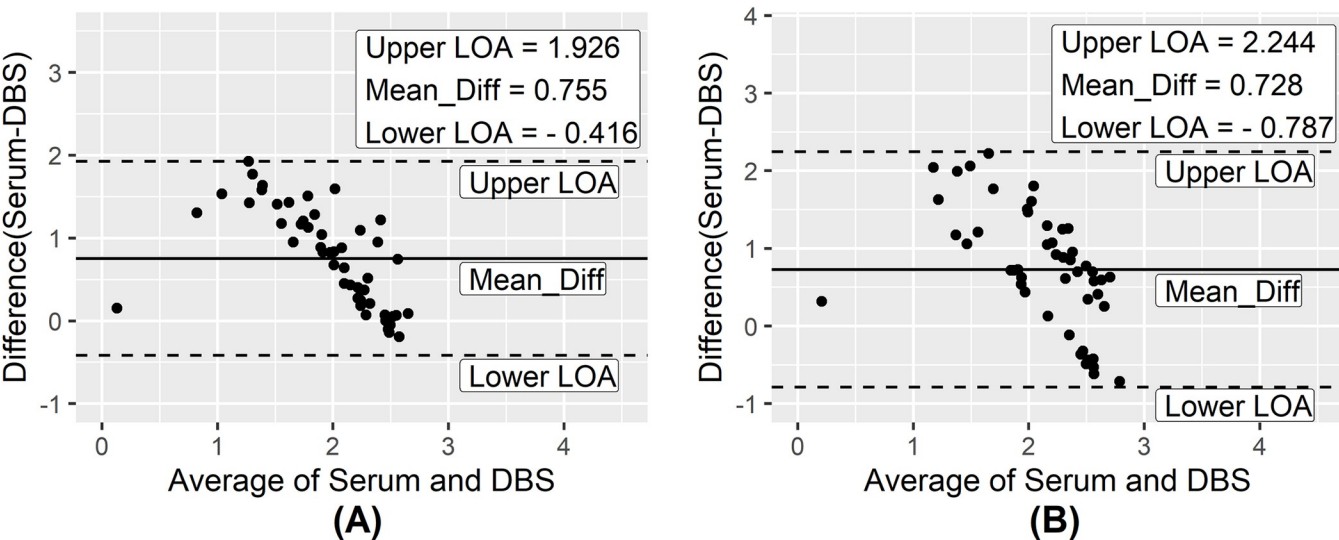

**Fig 6. Bland-Altman analysis to determine the agreement between serological assays (A: rK28 ELISA; B: rK39 ELISA) using different clinical samples (DBS–dried blood spots).**

**Table 5. Summary of the diagnostic performance of different assays using filter paper sampling technique compared to the current study.**

| Disease Investigated | Reference | Country | Study population | Filter paper used | Assay | Clinical specimen tested | Reference test | Sensitivity (95% CI) | Specificity (95% CI) |
|---|---|---|---|---|---|---|---|---|---|
| Cutaneous Leishmaniasis | Mota *et al* [9]. | Brazil | N = 48 (Suspected) | Whatman FTA card | PCR | Lesion scrapings | Direct Microscopy | 100 (85.7–100) | 92 (75–97.8) |
| | | | | | | | PCR (samples in STE buffer) | 100 (86.2–100) | 96 (79.7–99.3) |
| | Miranda *et al* [8]. | Panama | N = 100 (Suspected) | FTA Micro Cards | PCR | Lesion scrapings | Smears, culture and/or PCR (samples in TE) | 94 (86–98) | Not estimated |
| | Morais *et al* [10]. | Brazil | N = 124 (Cases) | Sterile regular filter paper | Duplex-qPCR | Lesion scrapings | Parasitological (microscopy and/or parasite isolation) and/or molecular test (conventional and/or qPCR targeting kDNA) | 83.06 | 86.96 |
| | Boggild *et al* [38]. | Peru | N = 45 (Suspected) | 7-cm filter paper | PCR | Lesion impressions | Any two of smear, aspirate culture, invasive-specimen PCR (scrapings and aspirates), filter paper PCR, and *Leishmania* skin test positive. | 92.3 (85.1–99.5) | 100 |
| Visceral Leishmaniasis | Hossain *et al* [28]. | Bangladesh | N = 80 (Cases) | WhatmanFTA cards | qPCR | Whole blood | Clinical evaluation and Rk39 rapid diagnostic test | 72.5 (61·38–81.90) | 100 (95.49–100) |
| | | | | | Loopamp | Whole blood | Clinical evaluation and Rk39 rapid diagnostic test | 85 (75·26–92·00) | 100 (95.49–100) |
| | Current study | Bangladesh | N = 50 (Cases) | Whatman FTA cards | qPCR | Whole blood | Clinical evaluation, Rk39 rapid diagnostic test and DAT | 76 (61.83–86.94) | 100 (92.89–100) |
| | | | | | RPA | Whole blood | Clinical evaluation, Rk39 rapid diagnostic test and DAT | 68 (53.30–80.48) | 100 (92.89–100) |
| | | | | | rK39 ELISA | Whole blood | Clinical evaluation, Rk39 rapid diagnostic test and DAT | 100 (92.89–100) | 100 (92.89–100) |
| | | | | | rK28 ELISA | Whole blood | Clinical evaluation, Rk39 rapid diagnostic test and DAT | 100 (92.89–100) | 100 (92.89–100) |

resources and tools is a valid approach. However, addressing the current needs demands a cost-effective and field feasible sampling method.

In recent times, DBS has been a popular surrogate sampling tool to whole blood for clinical studies. Relative to collection of whole blood, DBS offers a method with relatively easy sample collection by using finger pricks, and reduced transportation and storage concerns that render its implementation in mass screening programs far more attainable. It is worth mentioning that the FTA card being used in the current study has the advantages over the regular filter paper, as the former one helps fixing the nucleic acid on the filter substrate for better outcome in the downstream analysis. Therefore, in advance of application in mass screening, the current study evaluated if the DBS sampling method is suitable as an alternative to the whole blood sampling method currently used in the diagnosis of visceral leishmaniasis.

The most promising finding of the study was the absolute sensitivity and specificity of rK39 ELISA observed for both serum and DBS samples in detecting *Leishmania* infection. In our previous study, we observed optimum sensitivity of rK39 ELISA (>97%) in identifying VL cases and higher specificity (>97%) while using serum as clinical sample [19]. Another small study (n = 15) performed in Sudan, showed more than 90% sensitivity of rK39 ELISA to detect Kala-azar cases through utilizing filter paper elute [23]. Similarly Vaish *et al.* reported 100% sensitivity and specificity between 88.4%-97.9% of rK39 ELISA in serum which again corroborate the findings of the present study [24].

In our previous study, we showed rK28 ELISA sensitivity >98% and specificity >96% in identifying VL patient's serum [19]. The very first study on rK28 ELISA reported the sensitivity and specificity of >96% in detecting the antigen specific anti-leishmanial antibodies in serum [25]. With no exception, Vaish *et al* showed 99.6% sensitivity and a specificity between 94.17%-100% of rK28 which again support the findings of the present study [26]. For the first time, in this study, the performance of rK28 ELISA was determined using both serum and DBS samples to screen VL patients. Similar to rK39 ELISA, the rK28 ELISA showed high sensitivity of 98.00% and 100.00% for serum and DBS respectively, with each demonstrating 100% specificity.

Another striking finding of the study is the excellent diagnostic accuracy of both the rK28 and rK39 ELISA, with AUC ranging from 0.997 to 1.00 determined through ROC analysis. Similarly, a high degree (AUC:0.97–0.99) of diagnostic accuracy for both serum rK28 and rK39 ELISA was observed in our previous study [19]. Furthermore, concordant to our results another investigation also reported excellent performance and diagnostic accuracy (AUC: 0.995) of both rK28 and rK39 ELISA in serum through ROC curve analysis [26]. We extend those previous findings in this study by demonstrating, for the first time, significant agreement (ICC = 0.8–0.9) between the OD values of serum and DBS samples for both rK28 and rK39 ELISA. These data thereby assert the potential of DBS as an alternative to whole blood collection. This is consistent with previous reports for enteric diseases that found good to excellent agreement between OD values of serum and DBS samples in ELISA investigating antigen specific antibody responses [27]. Thus, although expected the demonstration that diagnostic performance of DBS is retained at levels comparable to those found with serum is an important step toward both wider access and mass screening.

In this study, qPCR performed with DNA extracted from whole blood showed a sensitivity of 94.0%, whereas our previous study showed 100% sensitivity in identifying VL cases from buffy coat DNA [2]. The minor discrepancy between the outcomes can be attributed to the variability of the clinical samples being used in the DNA extraction. In the earlier study, buffy coat was used which contains more *L. donovani* homing leukocytes compared with that in whole blood. While performing qPCR with DNA extracted from filter paper impregnated blood samples, 76.0% sensitivity was found which is congruent to our previous study where

72.5% sensitivity was demonstrated when DBS were used in qPCR assay [28]. In concordance with our previous studies, the qPCR assay showed 100% specificity for both DBS and whole blood in the current investigation [2,28]. The sensitivity of qPCR assay with dried blood sample is lower than that of whole blood because the lower abundance of the target DNA template for *L. donovani* in filter paper. Moreover, the higher Ct values of DBS compared to that of whole blood for the same VL cases reassure the lower amount of residual DNA of *Leishmania* in filter paper (Fig 4). In addition, most of the whole blood samples with higher Ct values (36–38) were found to be negative when the corresponding samples were DBS. In this study, the elusion volume (150 μL) of the filter paper extracted DNA was higher in respect to blood: extracted DNA ratio which could significantly impact the performance of the molecular assays. To overcome the challenge, elution of filter paper extracted DNA in a decreased volume might augment the efficiency of molecular assays.

In a study conducted in Northern Ethiopia Abbasi *et al.* reported promising analytical sensitivity of qPCR with DBS in the detection of *L. donovani*, however the assay efficiency depends on the abundance of parasite and the sample storage duration [29–30]. Along with VL, several studies have been conducted on cutaneous leishmaniasis to understand the usefulness of DBS for clinical applications. Numerous studies reported promising sensitivity and specificity of different molecular and serological assays with DBS in the diagnosis of VL and CL patients which favor its routine usage over other sampling techniques [7–14] (Table 5).

In the current study, we used an RPA assay for the first time to detect *L. donovani* DNA from DBS. The assay showed lower sensitivity of 68.0% with DBS which indicates the underperformance of the blood impregnated filter paper in detecting VL. As expected, the attributes for lower performance of real time PCR are applicable for the RPA assay while using the filter paper as the analysis matrix. Moreover, for most of the cases, higher TT values of DBS compared with that of whole blood reassure the lower amount of residual DNA of *L. donovani* in filter paper. In contrary, the RPA assay with whole blood showed a promising sensitivity of 92.0%, whereas our previous study demonstrated 100% sensitivity in detecting VL while using buffy coat as clinical specimens [18]. This decreased sensitivity of the RPA assay using whole blood also accounts for the lower amount of parasite DNA that was obtained from the whole blood compared to buffy coat. The specificity was 100% for both whole blood and DBS in the RPA assay. In the current study we found strong agreement (κ = 0.98) between qPCR and RPA in whole blood which is concordant with the findings of our previous study where RPA showed absolute agreement with qPCR [18]. Moreover, we showed promising agreement (κ = 0.735) between dried blood samples and whole blood in detecting leishmaniasis through qPCR which corroborate the findings of our previous study (κ = 0.784) [28].

The current study is presented with some limitations. The major limitation of this study lies in the relatively small sample size (50 VL cases versus 50 controls) which might influence the true clinical efficiency of the investigative methods. Irrespective of this, comparative analyses of paired serum and DBS samples was a key objective that was delivered, although that samples in the lower range of detection may have greater potential to reveal discrepancies. Thus, evaluation of a greater number of samples across the diagnostic range appears merited for diligence. At the post elimination period, a declining trend of VL cases is expected which will have impact on performing clinical study in a confined endemic setting. Such challenges can be overcome through inclusion of multiple endemic countries. An additional feature of the study design that likely does not replicate the ultimate use scenario is that the DBS samples used in this study were archived in -80˚C. Although preservation of DBS for a long period can harm the DNA, numerous studies have demonstrated that long term preservation in -20˚C or -80˚C does not significantly impact results, with quality and quantity of the DNA still considered adequate for clinical analysis [31–32]. All the healthy controls enrolled in this study were DAT

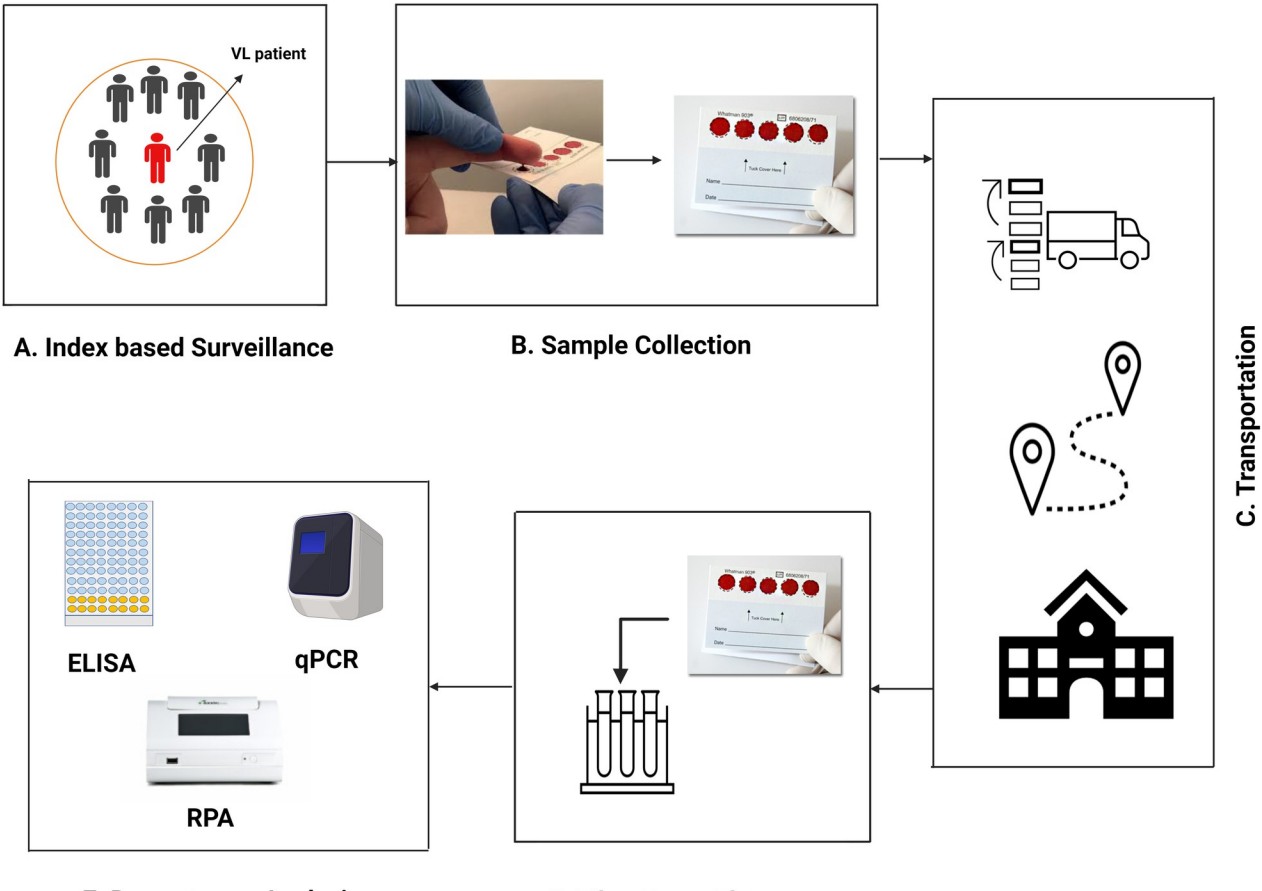

**Fig 7. Index-based surveillance to gauge residual *Leishmania* infection using a filter paper blood sampling method.** (A) Identification of index VL case and selection of its contacts for screening. (B) Collection of blood on filter paper and drying blood spots [39]. (C) Transport of filter papers with samples to the central facility. (D) Suitable buffer is applied to elute the analytes/biomarkers from filter paper. (E) Downstream analysis (ELISA, qPCR or RPA etc.) is performed to identify *Leishmania* infection. [The figure was generated in paid BioRender].

negative which may overestimate the efficiency of the assays. However, the laboratory personnel were blinded to the ID and types of cases and controls they were evaluating to nullify any operator bias. Moreover, tests were performed in duplicate to increase the reliability of the assays.

Our findings support the use of DBS as a minimally invasive technique for blood sample collection/transfer prior to performing downstream laboratory analyses. Together with the promising diagnostic efficiency of serological methods, this field feasible sampling strategy can be applied for mass screening in the endemic region during the post elimination period. Since the NKEP is stepping into the maintenance phase, formulation of an effective surveillance strategy is prerequisite to identify the residual infection that has been the key driver in inter-epidemic disease transmission. Moreover, an effective surveillance system is essential for WHO validation/certification of VL elimination. Towards addressing such endgame challenges, an index-based surveillance approach has been proposed and implemented in endemic settings for lymphatic filariasis (LF) together with an intervention package to curb transmission [33–34]. Moreover, our previous study, that implemented an index-based approach to explore the association between asymptomatic VL infection with post–kala-azar dermal leishmaniasis (PKDL) patients, provided further evidence to support this pragmatic approach for

gauging residual infection since VL/PKDL transmission is highly clustered [35]. In the peri and post elimination period surveillance of the asymptomatic carriers might be challenging as there is no defined marker for this these cases. To date most of the studies defined such cases with the serological markers including rK-39 RDT, rK-39 ELISA etc. A recent study conducted by Mondal et al. showed that a certain titer of rK-39 antibody can predict asymptomatic infection progression to VL within 24 months by 87.5% sensitivity and 95% specificity [36]. This encouraging finding is in favor to adopt such serological methods coupled with dried blood spots at post elimination period for detecting the residual infection and monitoring their progression to VL case. In addition, Hollie-Ann Hatherell *et al* emphasized the application of DBS for sustainable surveillance of NTDs for the post-elimination era [37]. Thus, in the light of earlier findings and propositions, we have formulated (Fig 7) an approach using the DBS for mass surveillance at post-elimination period of VL.

In the wake of multitude advantages of the DBS technique including the time and equipment needed for collection and the ease of processing, storage and shipment, its routine usage in clinical applications has been soaring. Another notable advantage in using DBS lies in the reduction of shipment costs compared to blood/serum/plasma collection which necessitates shipment of the samples while maintaining a costly cold chain process. A previous study reported the estimated shipment cost of $75–285 for a 10-pound package of dry ice which can be avoided with DBS cards [31]. However, further methodological improvement is warranted to increase the DNA yield from filter paper to perform molecular methods. We envisage that the above findings and recommendations will help policy makers to adopt an effective mass surveillance strategy towards expediting the elimination of VL from the endemic regions and its eventual eradication. Finally, a future large-scale prospective study should be performed to discern the proper utility of this sampling technique coupled with the proposed serological assays.

## Supporting information

**S1 Table. The primers and probe sequences of real time PCR used in the study.** (DOCX)

**S2 Table. The primers and probe sequences of RPA assay used in the study.** (DOCX)

## Acknowledgments

We are grateful to all of the participants for their valuable participation in this study. The authors are thankful to core donors include the Government of the People's Republic of Bangladesh; Global Affairs Canada (GAC), Canada; Swedish International Development Cooperation Agency (Sida); and the Department for International Development (UKAid) for providing unrestricted support and commitment to icddr,b's research efforts.

## Author Contributions

**Conceptualization:** Prakash Ghosh, Rajashree Chowdhury, Md. Abu Rahat.

**Data curation:** Prakash Ghosh, Rajashree Chowdhury, Md. Abu Rahat, Nur E Arpha.

**Formal analysis:** Prakash Ghosh, Rajashree Chowdhury, Md. Abu Rahat.

**Funding acquisition:** Prakash Ghosh.

**Investigation:** Rajashree Chowdhury, Md. Abu Rahat, Faria Hossain, Nur E Arpha.

**Methodology:** Prakash Ghosh, Rajashree Chowdhury, Md. Abu Rahat, Yasuyuki Goto, Malcom S. Duthie.

**Project administration:** Prakash Ghosh, Rajashree Chowdhury, Md. Abu Rahat.

**Resources:** Prakash Ghosh, Rajashree Chowdhury, Md. Abu Rahat.

**Software:** Prakash Ghosh, Rajashree Chowdhury, Md. Abu Rahat.

**Supervision:** Ahmed Abd El Wahed, Susana Campino, Mary Cameron, Rashidul Haque, Dinesh Mondal.

**Validation:** Prakash Ghosh, Rajashree Chowdhury, Md. Abu Rahat, Faria Hossain, Mojca Kristan, Matthew Higgins, M. M. Towhidul Islam.

**Visualization:** Prakash Ghosh, Rajashree Chowdhury, Md. Abu Rahat.

**Writing – original draft:** Prakash Ghosh, Rajashree Chowdhury, Md. Abu Rahat.

**Writing – review & editing:** Mojca Kristan, Matthew Higgins, Ahmed Abd El Wahed, Yasuyuki Goto, M. M. Towhidul Islam, Susana Campino, Mary Cameron, Malcom S. Duthie, Rashidul Haque, Dinesh Mondal.

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
