## [Decision Letter · Decision Letter 0]

12 Jul 2023

Dear Dr. Mondal,

Thank you very much for submitting your manuscript "Dried Blood Spots (DBS): a suitable alternative to using whole blood samples for diagnostic testing of visceral leishmaniasis in the post-elimination era" for consideration at PLOS Neglected Tropical Diseases. As with all papers reviewed by the journal, your manuscript was reviewed by members of the editorial board and by several independent reviewers. The reviewers appreciated the attention to an important topic. Based on the reviews, we are likely to accept this manuscript for publication, providing that you modify the manuscript according to the review recommendations. 

Sincerely,

Travis J Bourret

Academic Editor

Walderez Dutra

Section Editor

Reviewer's Responses to Questions

**Key Review Criteria Required for Acceptance?**

**Methods**

-Are the objectives of the study clearly articulated with a clear testable hypothesis stated?

-Is the study design appropriate to address the stated objectives?

-Is the population clearly described and appropriate for the hypothesis being tested?

-Is the sample size sufficient to ensure adequate power to address the hypothesis being tested?

-Were correct statistical analysis used to support conclusions?

-Are there concerns about ethical or regulatory requirements being met?

Reviewer #1: The hypothesis that continued surveillance must be done to preserve the gains made by the elimination campaign is widely accepted in the VL research community. However, the insistence that serology will be a good indicator of any 

emerging foci needs to be clarified. Anti-rK39 and rK28 antibodies are found at high levels in South Asian endemic regions, compared to East Africa and are long lasting. Hence sero-surveys though useful in assessing endemicity, may not consistent predictors of new infections, which is the key need post- elimination. PCR may be a better predictor of that. These points need to be considered in the rewriting of the draft.

Reviewer #2: Methodology part need some improvements as suggested. The statement of invasive needs to change with appropriate word as DBS collection is as invasive as fresh sample collection i. e. use of needle.

**Results**

-Does the analysis presented match the analysis plan?

-Are the results clearly and completely presented?

-Are the figures (Tables, Images) of sufficient quality for clarity?

Reviewer #1: The results and analyses and their representations are sufficient for the serological tests. Statistical methods are appropriate. However for the PCR methods, more details must be include din the results section for clarity and completion.

Reviewer #2: Result is OK. P value of fig. 3 may be reevaluated.

**Conclusions**

-Are the conclusions supported by the data presented?

-Are the limitations of analysis clearly described?

-Do the authors discuss how these data can be helpful to advance our understanding of the topic under study?

-Is public health relevance addressed?

Reviewer #1: The conclusions are in line with the results mostly. However, the limitations of the analysis must be more clearly discussed. The sample size is mentioned, and the inclusion of a larger sample size is alluded to. This is to be expected. But, given the success of the elimination campaign, it is highly likely that larger a larger cohort of VL patients is not available. The authors must mention this and how it will impact any future studies. 

Apart form that, the authors must also clearly state the advantages and dis advantages of using serology form surveillance in endemic foci. This has been a subject of extensive discussion in the VL field and how other techniques can fill the gap, namely PCR and antigen detection urine tests. Ihat is a clear limitation of the study and DBS utility, which has not been discussed in this manuscript.

Reviewer #2: Conclusion is well discussed but the importance of this study is not as much important as claimed. Need to address the point

**Editorial and Data Presentation Modifications?**

Reviewer #1: The authors must include the discussion on the limitations of sero surveillance that are suggested above in the revised manuscript.

The authors must include the Limit of detection of both the qPCR and RPA-PCR for whole blood and DBS in the results section. They only disclose the Ct and TT values, not the absolute levels of parasite genome equivalents or DNA concentrations that are detected by the methods. It is also necessary that the authors disclose the yield of DNA from DBS, and discuss its correlation to whole blood, since it is a highly likely contributor to the differences in sensitivity. 

These details must be included in the revised manuscript.

Reviewer #2: 1. Which Whatman FTA card was used? There are various categories.

2. What was the basis of using two 5mm circles to elute DBS in 150 and 200 micro lit. elution buffer? Further, how the authors achieved 1:2000 dilutions from DBS? Previously Harith et al. 1989, Bimal et al. 2005 and many others had eluted single 5mm circle in 165 micro lit. elution buffer to get 1:50 dilution of antibodies. Quantification may be explained in the text.

3. After how long the samples collected were eluted? It also may be indicated in figure 7.

4. The sample was stored at -20oC till further use. How long it was stored? Prolonged storage in -20 deg. C loose Ab titer and protein. Definitely there is a value of deterioration. So, actual storage time must be recorded in methodology.

5. Statistical significance may be rechecked by an statistician for Fig. 3 

6. Primers used may be provided in the text for consistency?

7. Invasive statement may be changed/ corrected. Isolation of DBS is not non invasive.

8. Fig. 1 and 2 must be reuploaded with good resolution and background.

**Summary and General Comments**

Reviewer #1: DBS is very promising for sero-surveillance and has been successfully implemented for other NTDSs such as filariasis that are also part of elimination campaigns. This is the first instance of a systematic study of DBS utility in the VL elimination campaign of South Asia. The authors have used the study to state the utility of DBS in Bangladesh clearly and precisely. The general execution, ethics, statistics and writing are very good and the authors clearly state the hypotheses and results. With the above suggested minor revisions, the manuscript is suitable for publication.

Reviewer #2: The manuscript “Dried Blood Spots (DBS): a suitable alternative to using whole blood samples for diagnostic testing of visceral leishmaniasis in the post-elimination era” has compared dried blood samples in comparison to whole blood NA for qPCR and RPA and serum for ELISA. Dried blood samples are being used to diagnose Leishmaniasis for years. In many areas, it was the sample of choice to diagnose Leishmaniasis because of comfortable transportation and storage. It was used mostly for sero-diagnosis (DAT and ELISA) and various molecular diagnostics. But now in the presence of rapid diagnostic kit; importance of DBS is not as much because the former needs only one drop blood from finger puncture and result comes within few minutes and don’t need much expertise and lab wares. Hence importance of this method is not as much in post RDK/RDT. However, it is a good effort to keep an alternative ready for the situations like non availability of RDK/RDT on the spot. Since the data shows compromised result in qPCR and RPA, the DBS cannot be an alternative of fresh blood sample. The proposed system is also as invasive as fresh blood collection.

PLOS authors have the option to publish the peer review history of their article (what does this mean?). If published, this will include your full peer review and any attached files.

Reviewer #1: Yes: Aarthy C Vallur

Reviewer #2: Yes: Dr. Shubhankar Kumar Singh

Figure Files:

Data Requirements:

Reproducibility:

References

---

## [Decision Letter · Decision Letter 1]

25 Sep 2023

Dear Dr. Mondal,

We are pleased to inform you that your manuscript 'Dried Blood Spots (DBS): a suitable alternative to using whole blood samples for diagnostic testing of visceral leishmaniasis in the post-elimination era' has been provisionally accepted for publication in PLOS Neglected Tropical Diseases.

Best regards,

Travis J Bourret

Academic Editor

Walderez Dutra

Section Editor

Reviewer's Responses to Questions

**Key Review Criteria Required for Acceptance?**

**Methods**

-Are the objectives of the study clearly articulated with a clear testable hypothesis stated?

-Is the study design appropriate to address the stated objectives?

-Is the population clearly described and appropriate for the hypothesis being tested?

-Is the sample size sufficient to ensure adequate power to address the hypothesis being tested?

-Were correct statistical analysis used to support conclusions?

-Are there concerns about ethical or regulatory requirements being met?

Reviewer #1: With the current revisions, the manuscript is acceptable for publication. The sample size question has been addressed by the authors along with the suitability of serology in this population.

Reviewer #2: -Are the objectives of the study clearly articulated with a clear testable hypothesis stated? Yes

-Is the study design appropriate to address the stated objectives? Yes

-Is the population clearly described and appropriate for the hypothesis being tested? Yes

-Is the sample size sufficient to ensure adequate power to address the hypothesis being tested? Yes

-Were correct statistical analysis used to support conclusions? Yes

-Are there concerns about ethical or regulatory requirements being met? Yes

**Results**

-Does the analysis presented match the analysis plan?

-Are the results clearly and completely presented?

-Are the figures (Tables, Images) of sufficient quality for clarity?

Reviewer #1: Yes, the analysis results and figures are appropriate. High- res figures are now included.

Reviewer #2: -Does the analysis presented match the analysis plan? Yes

-Are the results clearly and completely presented? Yes

-Are the figures (Tables, Images) of sufficient quality for clarity? Yes

**Conclusions**

-Are the conclusions supported by the data presented?

-Are the limitations of analysis clearly described?

-Do the authors discuss how these data can be helpful to advance our understanding of the topic under study?

-Is public health relevance addressed?

Reviewer #1: The conclusions hinge on their observations in this study and previous studies that have been used to support the use of DSBs. They are sufficient to justify the premise.

Reviewer #2: -Are the conclusions supported by the data presented? Yes

-Are the limitations of analysis clearly described? Yes

-Do the authors discuss how these data can be helpful to advance our understanding of the topic under study? Yes

-Is public health relevance addressed? Yes

**Editorial and Data Presentation Modifications?**

Reviewer #1: Accept for publication

Reviewer #2: None

**Summary and General Comments**

Reviewer #1: DBS is very promising for sero-surveillance and has been successfully implemented

for other NTDSs such as filariasis that are also part of elimination campaigns. This is the first

instance of a systematic study of DBS utility in the VL elimination campaign of South Asia. The

authors have used the study to state the utility of DBS in Bangladesh clearly and precisely. The

general execution, ethics, statistics and writing are very good and the authors clearly state the

hypotheses and results.

Reviewer #2: (No Response)

PLOS authors have the option to publish the peer review history of their article (what does this mean?). If published, this will include your full peer review and any attached files.

Reviewer #1: No

Reviewer #2: **Yes: **Dr. Shubhankar Kumar Singh

---

## [Editor Report · Acceptance letter]

6 Oct 2023

Dear Dr. Mondal,

We are delighted to inform you that your manuscript, "Dried Blood Spots (DBS): a suitable alternative to using whole blood samples for diagnostic testing of visceral leishmaniasis in the post-elimination era," has been formally accepted for publication in PLOS Neglected Tropical Diseases.

Best regards,

Shaden Kamhawi

co-Editor-in-Chief

Paul Brindley

co-Editor-in-Chief
